# First Nation-Wide Study of the Incidence and Characteristics of Retinal Detachment in Poland during 2013–2019

**DOI:** 10.3390/jcm12041461

**Published:** 2023-02-12

**Authors:** Michal Szymon Nowak, Michał Żurek, Iwona Grabska-Liberek, Piotr Kanclerz

**Affiliations:** 1Institute of Optics and Optometry, University of Social Science, 121 Gdanska Str., 90-519 Lodz, Poland; 2Provisus Eye Clinic, 112 Redzinska Str., 42-209 Czestochowa, Poland; 3Doctoral School, Medical University of Warsaw, 61 Zwirki and Wigury Str., 02-091 Warsaw, Poland; 4Department of Analyses and Strategies, Ministry of Health, 15 Miodowa Str., 00-952 Warsaw, Poland; 5Department of Ophthalmology, Center of Postgraduate Medical Education, 231 Czerniakowska Str., 01-416 Warsaw, Poland; 6Hygeia Medical Clinic, Department of Ophthalmology, 80-286 Gdansk, Poland; 7Helsinki Retina Research Group, University of Helsinki, 00014 Helsinki, Finland

**Keywords:** rhegmatogenous RD, traction RD, serous RD

## Abstract

Aims: The present study aimed to analyze the incidence and characteristics of all types of retinal detachment (RD) in the overall population of Polish adults during 2013–2019. Methods: Data from all levels of healthcare services at public and private institutions recorded in the National Health Fund (NHF) database were evaluated. International Classification of Diseases codes (ICD-9 and ICD-10) and unique NHF codes were used to identify RD patients and RD treatment procedures. Results: In the period 2013–2019, 71,073 patients with RD were newly diagnosed in Poland. The average incidence was 32.64/100,000 person-years (95% CI: 31.28–33.99) and it increased with the age of patients, with the highest rate in the group of patients ≥70 years of age. The overall incidences of rhegmatogenous RD, traction RD, serous RD, other RD and unspecified RD were 13.72/100,000, 2.03/100,000, 1.02/100,000, 7.90/100,000 and 7.97/100,000 person-years, respectively. The most common surgical treatment for RD in Poland was PPV performed on average in 49.80% of RD patients. The risk factor analyses showed that rhegmatogenous RD was significantly associated with age (OR 1.026), male sex (OR 2.320), rural residence (OR 0.958), DM type 2 (OR 1.603), any DR (OR 2.109), myopia (OR 2.997), glaucoma (OR 2.169) and uveitis (OR 2.561). Traction RD was also significantly associated with age (OR 1.013) and male sex (OR 2.785) as well as with any DR (OR 2.493), myopia (OR 2.255), glaucoma (OR 1.904) and uveitis (OR 4.214). Serous RD was significantly associated with all analyzed risk factors except DM type 2. Conclusions: The total incidence of retinal detachment in Poland was higher than found in previously published studies. Our study demonstrated that diabetes type 1 and diabetic retinopathy are risk factors of development of serous RD, which is presumably associated with the disruption of the blood–retinal barriers in these conditions.

## 1. Introduction

Retinal detachment (RD) is a critical, potentially blinding eye condition; however, there are only a few population-based studies reporting the incidence of RD in different parts of the world and data studied are either old or limited to rhegmatogenous RD only.

In the United States, the annual incidence of RD was approximately 12 per 100,000 population [1,2]. In Europe, the annual incidence of 4 per 100,000 population was reported in Switzerland [3], 6.9 in Finland [4] and 10.6 per 100,000 in Sweden [5]. In Asia, the average RD incidence in Singapore was 10.5 per 100,000 population; the age-adjusted relative risk of an RD surgery in Chinese was 3.0 (95% CI: 2.9–3.1) compared with Indians [6].

The most frequent form of RD is rhegmatogenous RD, in which a retinal tear allows liquefied vitreous humor, commonly due to degeneration of the vitreous body. The annual reported incidence of rhegmatogenous RD varied from 2.6 per 100,000 in France during the severe acute respiratory syndrome coronavirus-2 pandemic (COVID-19) [7], 9.5 to 12.1 per 100,000 in the United Kingdom [8,9], 13.7 per 100,000 in Denmark [10], 14.0 per 100,000 in Sweden [11] to 18.2 per 100,000 in the Netherlands [12]. Asians may have lower incidence of rhegmatogenous RD than Caucasians. [8]. Moreover, nontraumatic phakic detachments had the highest incidence of 9.7 per 100,000, while nontraumatic pseudophakic and aphakic RD were significantly less common, at 1.2 and 0.3 per 100,000, respectively [8]. 

The present study aimed to analyze the incidence and characteristics of all types of retinal detachment in the overall population of Polish adults in the period 2013–2019 and to report the co-existing risk factors. To the best of our knowledge, this is the first study aimed at searching the incidence of all types of retinal detachment in the nation-wide population in twenty first century.

## 2. Materials and Methods

### 2.1. Data Sources, Disease Codes and Definitions

The present study was part of the project “Maps of Healthcare Needs—Database of Systemic and Implementation Analyses” and was co-financed by the European Union funds through the European Social Fund under the Knowledge Education and Development Operational Program (EU grant number: POWR 05.02.00-00-0149/15-01). The Polish Ministry of Health, which is entitled by the Law of Republic of Poland to process the data of the national database of hospitalization, approved the study protocol. However, we did not need to obtain ethics committee approval; the study adhered to the tenets of the Declaration of Helsinki for research involving human subjects. The study design was a retrospective and nationwide survey which was described in detail in our previous papers [13,14,15,16,17,18,19]. The data of all adult patients who were diagnosed with retinal detachment between 1 January 2013 and 31 December 2019 were extracted from the national database of hospitalization [20]. This database is maintained by the National Health Fund (NHF) and records all medical procedures in public and private hospitals in Poland financed from public sources. It provides accurate population-based medical data which include the diagnoses coded according to the International Classification of Diseases, 10th Revision (ICD-10), and all procedures performed were coded using the International Classification of Diseases, 9th Revision (ICD-9) procedure codes and unique NHF codes corresponding to certain hospital procedures. It also compiles demographical features such as personal identification number (PESEL), date of birth, area code, and sex of patients. Data regarding the population of Poland were obtained from the Central Statistical Office of Poland [21].

During the study period, in the years 2013–2019, each individual patient with retinal detachment in the national database of hospitalization was identified with ICD-10 codes H33.0, H33.2, H33.4, H33.5 and H33 (unspecified RD). The ICD-9 codes 14.3 with extensions were used to identify repair of retinal tear with diathermy, cryotherapy or laser photocoagulation. The ICD-9 codes 14.4 with extensions were used to identify repair of retinal detachment with scleral buckling and implant. The ICD-9 codes 14.5 with extensions were used to identify repair of retinal detachment with diathermy, cryotherapy or laser photocoagulation. The ICD-9 codes 14.74 and 14.75 were used to identify repair of retinal detachment with pars plana vitrectomy (and injection of vitreous substitute). Other treatment of retinal detachment was identified with ICD-9 code 14.9.

### 2.2. Statistical Analyses

In the first part of the study, the descriptive statistics of the retinal detachment incidence during 2013–2019 were performed. The incidence of RD was presented for each year separately and by age category matched with the corresponding year population data in Poland. Other statistical analyses also included the demographic characteristics of patients with RD and particular comorbid diseases. The socio-demographic data of patients including age, sex and place of residence were anonymously recorded. The study flowchart is presented in Figure 1. The second part of the present study included clinical and treatment characteristics of RD and risk factor analysis. The number of patients having RD with retinal break, serous RD, traction RD, other and unspecified RDs were collected. Statistical analysis also included calculations of RD treatments by different surgery methods. Multiple logistic regression was used to investigate the association of different types of RD with several risk factors including age, gender, rural residence, diabetes mellitus (DM) type 1 and 2, diabetes retinopathy (DR), myopia, glaucoma and uveitis. For regression analysis, the adequate control group of patients was randomly selected from the NHF database from patients treated in 2013–2019 who had not been diagnosed with RD. To eliminate the problem of strong collinearity, the effect of VIF coefficients, Pearson’s linear correlation coefficients and Spearman’s nonlinear correlation were checked. Odds ratios (ORs) were computed and *p* values less than 0.05 were considered statistically significant. R statistical software V. 3.6.2 was used for all analyses. 

## 3. Results

In the period 2013–2019, 71,073 patients with RD were newly diagnosed in Poland and 10,897 patients were diagnosed earlier, but required reoperation during the study period (Figure 1). The average incidence per 100,000 person-years was 32.64 (95% CI: 31.28–33.99) and it increased with the age of patients, from 8.63 in the 19–29 age group to 73.41 in the group of patients ≥70 years of age. The incidence was slightly higher in men when compared to women and the sex distribution differed between RD subtypes. Detailed results are presented in Table 1.

The age analysis of patients did not show any particular trend in the period 2013–2019 and the mean age of patients was 59.82 ± 14.51 years. The demographic characteristics of patients with RD and comorbid diseases are presented in Table 2. There was a slight majority of women in the study group (52.27%). The vast majority of patients were urban residents (68.02%). The analysis of comorbidities indicated a high proportion of glaucoma among RD patients—on average 32.73% of patients. This percentage has been systematically decreasing, from 37% in 2013 to 24.28% in 2019. In turn, the presence of pseudophakia was reported in 26.41% of patients. Among chronic diseases, DM type 2 was found in 26.04%, and DM type 1 in 9.89% of all included subjects. When analyzing the percentage of RD patients with both types of diabetes, downward trends were noticeable in the analyzed period (especially in 2014–2019). Other comorbidities included: diabetic retinopathy found in 6.26% of RD patients, myopia found in 13.26%, history of uveitis found in 3.54%, aphakia in 3.21% and lens displacement in 2.02%.

The clinical characteristic of patients with RD is presented in Table 3. The most common detailed diagnosis according to the ICD-10 classification in the analyzed group was RD with retinal tear (rhegmatogenous) (H33.0) which was reported in 42.03% of patients, followed by traction RD (H33.4) in 6.23% of patients and serous RD (H33.2) in 3.14% of patients. The other RD (H33.5) and unspecified RD (H33) cases were diagnosed in 24.19% and 24.40% of patients, respectively. The overall incidences of rhegmatogenous RD, traction RD, serous RD, other RD and unspecified RD were 13.72/100,000, 2.03/100,000, 1.02/100,000, 7.90/100,000 and 7.97/100,000 person-years, respectively. The results of the surgical treatment of RD revealed the most common type of surgery was mechanical vitrectomy and/or injection of a vitreous substitute (14.74 and 14.75 according to ICD-9 classification). This type of surgery was performed on average in 49.80% of RD patients, with the number of surgeries significantly increased in the study period (from 4312 to 5792 procedures per year); 7.57% of patients underwent a retinal detachment repair with scleral buckling and implant (ICD-9: 14.4), and the number of these procedures decreased from 1055 to 602 in the analyzed period. Other repair procedures of RD were performed in a small number of patients. However, during the study period 33.55% of all RD patients did not receive any treatment in Poland.

Multivariate logistic regression models were constructed to analyze the risk factors for rhegmatogenous RD (with retinal break), traction and serous RD in Poland during 2013–2019 (Table 4). Our analysis showed that rhegmatogenous RD was significantly associated with age (OR 1.026), male sex (OR 2.320), rural residence (OR 0.958), DM type 2 (OR 1.603), any DR (OR 2.109), myopia (OR 2.997), glaucoma (OR 2.169) and uveitis (OR 2.561). Traction RD was also significantly associated with age (OR 1.013) and male sex (OR 2.785) as well as with any DR (OR 2.493), myopia (OR 2.255), glaucoma (OR 1.904) and uveitis (OR 4.214). Serous RD was significantly associated with all analyzed risk factors except DM type 2.

## 4. Discussion

This is the first study presenting results of an extensive analysis of the incidence of RD in Poland. There have been single European studies evaluating the incidence of particular subtypes of RD and their risk factors which have been published in recent years. In Poland, during the study period, the average incidence of all types of RD was 32.6 per 100,000 person-years and it increased with the age of patients—from 8.63 in the 19–29 age group to 73.41 in the group of patients ≥70 years of age. The incidence was slightly higher in men when compared to women. The most common was rhegmatogenous RD (H33.0), which was reported in 42.03% of patients, followed by traction RD (H33.4) in 6.23% of patients and serous RD (H33.2) in the 3.14% of patients. The overall incidences of rhegmatogenous RD, traction RD and serous RD were 13.72/100,000, 2.03/100,000 and 1.02/100,000 person-years, respectively. The other RD (H33.5) and unspecified RD (H33) cases were diagnosed in 24.19% and 24.40% of patients, which may have biased the findings of specific subtypes. However, all included subjects were identified by matching personal identification number (PESEL) with the ICD-10 code, some of specific RDs (H33.0, H33.2, H33.4) could have been coded as H33.5 or H33 and we are not able to verify these. 

It is well known that rhegmatogenous RD is more common in males than in females [9]. In our study, male sex was a risk factor for all types of RD including rhegmatogenous, tractional and serous RDs (odds ratio 2.32 [95% CI: 2.26–2.38], 2.79 [95% CI: 2.64–2.94] and 2.34 [2.25–2.43], respectively). The incidence of RD was the highest in patients aged 70 years or more (age standardized incidence 73.4 vs. in all age groups 32.6). This is similar to what has been reported in the United Kingdom, where the peak age was the 70 to 79-year group (incidence of 29.1 per 100,000) in the Wolverhampton region, whereas in Walsall it was 98.6 per 100,000 in the 85+ age group [8]. Other studies have found a peak in both sexes in the 60- to 69-year age group [9]. Laatikainen et al. noted that the mean age of rhegmatogenous RD was 54.2 years [4]. In a study by Algvere et al., the mean age at surgery for women was 62.9 years (95% CI: 61.5–64.4), while it was 58.3 years for men (95% CI: 57.1–59.4) [11]. On the other hand, Hajari et al. found that in Denmark the incidence of rhegmatogenous RD was negatively correlated with age, and was most common at younger ages [10]. The greater age of patients undergoing RD surgery could be associated with older age during cataract surgery, which is known to be a risk factor for retinal detachment. In our study, uveitis was found to be a risk factor for all types of RD: not only for tractional and serous RD, but also for rhegmatogenous RD (odds ratio 4.21 [95% CI: 3.73–4.75], 4.46 [95% CI: 4.05–4.91] and 2.56 [2.37–2.76], respectively). A previous study by De Hoog et al. found that uveitis is a risk factor for rhegmatogenous RD, which occurred in 3.1% of patients [22]. The aforementioned prevalence was significantly higher than in a general population and the prognosis in uveitic rhegmatogenous RD was poor. 

Our study has demonstrated that diabetes type 1 is a risk factor for serous (exudative) RD (OR 2.933; 95% CI: 2.757 to 3.121; *p* < 0.001). In serous RD, fluid collects between the photoreceptors and retinal pigment epithelium due to the disruption of the integrity of the blood–retinal barriers (BRBs). Several inflammatory, infection, infiltrative, neoplastic, vascular and degenerative conditions inducing hypoxia-ischemia may be associated with BRB breakdown and development of serous RD [23]. The disturbance of the blood–retinal barrier is also known to lead to the development of diabetic macular edema and diabetic retinopathy; however, to date diabetes type 1 or diabetic retinopathy has not been described as a risk factor of serous RD [24]. The inner BRB is formed by tight junctions between capillary endothelial cells [25], while the outer BRB is formed by tight junctions of RPE cells [26]. Diabetes leads to vascular endothelial cell dysfunction, diminishes the anti-atherogenic role of the vascular endothelium and might result in microvascular complications [27]. However, the disruption of outer rather than inner BRB may play a prominent role in development of serous RD; where there is a breakdown in the outer BRB, the commonest clinical manifestation is a localized or more generalized serous RD, the extent of which is dependent upon the level of outer BRB breakdown [23]. Alteration of several biochemical pathways occurs in the retinal pigment epithelium in diabetic patients and outer BRB-specific leakage has been demonstrated in diabetic patients [28,29]. Finally, a recent review concluded that serous RD may develop as a consequence of any condition that violated the BRB [23]; in these terms our findings are justified. There is no evidence showing that diabetes type 1 more significantly influences the outer BRB than diabetes type 2, in which we did not find an association with serous RD (OR 0.984; 95% CI: 0.932 to 1.039; *p* = 0.567); however, the RPE function is directly reduced by elevated blood glucose levels, which could be greater in diabetes type 1 [30]. It is obvious that the BRBs are most affected in diabetic retinopathy, which justifies why the OR is the highest in this case (OR 7.046; 95% CI: 6.602 to 7.521; *p* < 0.001).

Rhegmatogenous RD develops from retinal breaks occurring at sites of firm adhesion during posterior vitreous detachment or from atrophic holes [31]. Most studies indicated a preponderance of men regarding the incidence of rhegmatogenous RD [10,32,33,34,35]; our study demonstrated similar results with the OR of 2.32 (95% CI: 2.265 to 2.376; *p* < 0.001) in males. Men are at higher risk for ocular trauma, which could contribute either acutely or remotely to rhegmatogenous RD formation [36]. However, male preponderance exists even after exclusion of trauma and pseudophakic RD, which is more common in women [37]. This is also presumably associated with the fact that males have a longer axial length, more posteriorly located vitreous base and are more affected by posterior vitreous detachment-associated rhegmatogenous RD [38,39]. Vitreomacular adhesions have also been shown to be more robust in patients with diabetes [40,41]. Diabetes alters the magnitude of attachment of the vitreous gel to the macula, resulting in a solid and longer lasting attachment of the gel throughout the whole life [41]. This could explain why diabetes type 2 and diabetic retinopathy were shown to be risk factors for rhegmatogenous RD in our study (OR 1.603; 95% CI: 1.545 to 1.663; *p* < 0.001, and OR 2.109; 95% CI: 1.973 to 2.254; *p* < 0.001, respectively).

The most common surgical treatment of RD in Poland was mechanical vitrectomy and/or injection of a vitreous substitute performed on average in 49.80% of RD patients, followed by RD repair with scleral buckling and implant in 7.57% of patients. Other repair procedures of RD were performed in a small number of patients. However, our study did not cover the period of the COVID 19 pandemic; a decrease in the number of rhegmatogenous RD procedures by 41.6% during the first 8-week lockdown period in 2020 when compared to 2019 was observed in France [7].

Limitations of the present study include possible presence of misclassification i.e., errors in using specific ICD-9 and ICD-10 codes might have occurred at different levels of healthcare institutions (operating theaters, hospitals, NFZ offices). We believe, such mistakes likely had only a minor impact on the study findings. The population size, national recruitment and impact of its findings on public health are the most important strengths of the current study.

## 5. Conclusions

To the best of our knowledge, this is the first nation-wide study aimed at searching the incidence of all types of retinal detachment in twenty first century. The total incidence of retinal detachment in Poland was significantly higher than found in previously published studies. Our study demonstrated that diabetes type 1 and diabetic retinopathy are risk factors for the development of serous RD, which is presumably associated with the disruption of the BRBs in these conditions. Myopia and male sex have been known to be demographic risk factors for rhegmatogenous RD; our study has also demonstrated an association with glaucoma, uveitis, diabetes type 2 and diabetic retinopathy.

## Figures and Tables

**Figure 1 jcm-12-01461-f001:**
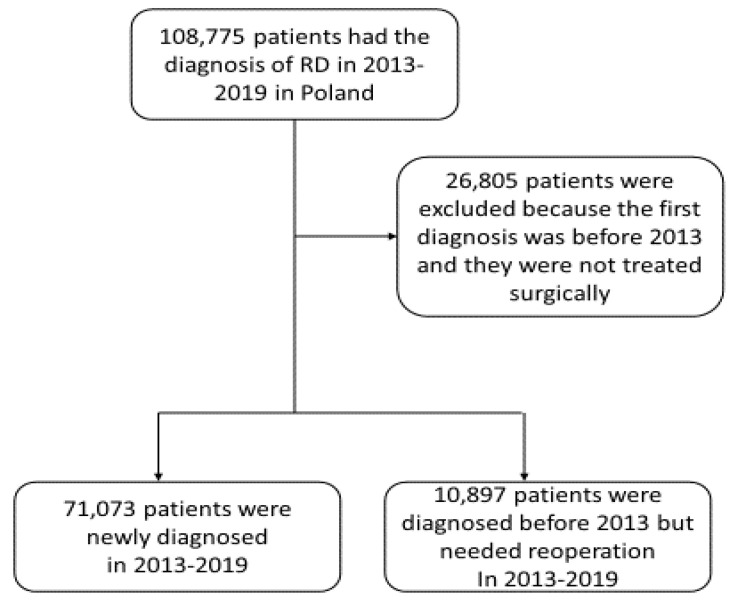
The study flowchart.

**Table 1 jcm-12-01461-t001:** Age-standardized incidence of RD among Polish adults from 2013 to 2019 by age group.

	2013	2014	2015	2016	2017	2018	2019	All
No. age 19–29 years (in thousands)	6116	5887	5667	5469	5280	5091	4917	38,427
No. of RD	483	479	552	497	427	458	421	3317
Incidence/100,000 person-yrs	7.90	8.14	9.74	9.09	8.09	8.99	8.56	8.63
Sex.% Women	42.24	43.22	46.38	45.47	43.79	41.92	46.08	44.2
No. age 30–39 years (in thousands)	6239	6314	6348	6331	6290	6235	6145	43,903
No. of RD	674	625	716	674	623	678	604	4594
Incidence/100,000 person-yrs	10.80	9.90	11.28	10.65	9.90	10.87	9.83	10.46
Sex.% Women	40.06	44	45.95	45.1	44.78	46.31	40.4	43.86
No. age 40–49 years (in thousands)	4880	4956	5065	5202	5342	5482	5632	36,558
No. of RD	771	808	827	920	868	944	990	6128
Incidence/100,000 person-yrs	15.80	16.30	16.33	17.69	16.25	17.22	17.58	16.76
Sex.% Women	43.58	40.35	43.17	42.39	45.05	44.07	44.24	43.31
No. age 50–59 years (in thousands)	5536	5406	5245	5089	4928	4783	4670	35,658
No. of RD	1906	1843	2011	1905	1826	1742	1701	12,934
Incidence/100,000 person-yrs	34.43	34.09	38.34	37.43	37.05	36.42	36.42	36.27
Sex.% Women	50	48.51	50.82	48.87	48.63	48.51	46.33	48.87
No. age 60–69 years (in thousands)	4410	4643	4888	5025	5127	5189	5219	34,501
No. of RD	2885	2974	3453	3409	3330	3546	3432	23,029
Incidence/100,000 person-yrs	65.42	64.05	70.64	67.84	64.95	68.34	65.76	66.75
Sex.% Women	55.77	52.35	55.26	53.74	54.8	53.27	50.38	53.62
No. age ≥70 years (in thousands)	3883	3905	3915	4031	4166	4319	4485	28,703
No. of RD	2788	2809	3001	3115	2995	3142	3221	21,071
Incidence/100,000 person-yrs	71.80	71.93	76.65	77.28	71.89	72.75	71.82	73.41
Sex.% Women	59.9	58.6	59.78	58.81	58.63	57	57.59	58.58
No. of all (in thousands)	31,064	31,112	31,128	31,147	31,134	31,100	31,068	21,7751
No. of RD	9507	9538	10,560	10,520	10,069	10,510	10,369	71,073
Incidence/100,000 person-yrs	30.60	30.66	33.92	33.77	32.34	33.79	33.37	32.64
Sex.% Women	53.03	51.43	53.66	52.42	52.9	51.83	50.61	52.27

Overall incidence/100,000 person-yrs–(95% CI:) 32.64 (95% CI: 31.28–33.99); Incidence in females/100,000 person-yrs–(95% CI:) 32.61 (95% CI: 31.11–34.11); Incidence in males/100,000 person-yrs–(95% CI:) 32.67 (95% CI: 31.16–34.12).

**Table 2 jcm-12-01461-t002:** The demographic characteristics of patients with RD and comorbid diseases.

	2013	2014	2015	2016	2017	2018	2019	All
Age mean ± SE	59.03 ± 14.73	59.26 ± 14.61	59.77 ± 14.59	60.01 ± 14.57	60.07 ± 14.37	59.9 ± 14.44	60.69 ± 14.27	59.82 ± 14.51
Women (*n*,%)	5042	4905	5666	5515	5326	5447	5248	37,149
	53.03	51.43	53.66	52.42	52.9	51.83	50.61	52.27
Men (*n*,%)	4465	4633	4894	5005	4743	5063	5121	33,924
	46.97	48.57	46.34	47.58	47.1	48.17	49.39	47.73
Urban residence (*n*,%)	6491	6531	7279	7168	6816	7118	6942	48,345
	68.28	68.47	68.93	68.14	67.69	67.73	66.95	68.02
Rural Residence (*n*,%)	3016	3007	3281	3352	3253	3392	3427	22,728
	31.72	31.53	31.07	31.86	32.31	32.27	33.05	31.98
DM type 1	1027	1054	1070	1039	931	1010	901	7032
E10	10.8	11.05	10.13	9.88	9.25	9.61	8.69	9.89
DM type 2	2615	2641	2808	2790	2575	2647	2428	18,504
E11	27.51	27.69	26.59	26.52	25.57	25.19	23.42	26.04
DR	649	679	677	641	586	660	559	4451
H36.0	6.83	7.12	6.41	6.09	5.82	6.28	5.39	6.26
Myopia	1207	1295	1415	1383	1422	1428	1274	9424
H44.2 and H52.1	12.7	13.58	13.4	13.15	14.12	13.59	12.29	13.26
Glaucoma	3511	3593	3518	3582	3269	3129	2500	23,102
H40	36.93	37.67	33.31	34.05	32.47	29.77	24.11	32.5
Uveitis	431	423	378	376	351	301	253	2513
H20	4.53	4.43	3.58	3.57	3.49	2.86	2.44	3.54
Pseudophakia	2531	2716	2812	2823	2720	2840	2327	18,769
Z96.1	26.62	28.48	26.63	26.83	27.01	27.02	22.44	26.41
Aphakia	323	363	346	353	315	312	268	2280
H27.0	3.4	3.81	3.28	3.36	3.13	2.97	2.58	3.21
Lens luxation	198	222	210	189	170	227	221	1437
H27.1	2.08	2.33	1.99	1.8	1.69	2.16	2.13	2.02

**Table 3 jcm-12-01461-t003:** The clinical characteristics of patients with RD in Poland during 2013–2019.

	2013	2014	2015	2016	2017	2018	2019	All
RD with retinal break *n* (%) H33.0	4127	4371	4072	4144	4185	4520	4455	29,874
	43.41	45.83	38.56	39.39	41.56	43.01	42.96	42.03
Serous RD *n* (%)	365	347	301	325	291	296	309	2234
H33.2	3.84	3.64	2.85	3.09	2.89	2.82	2.98	3.14
Traction RD *n* (%)	782	786	726	605	526	546	457	4428
H33.4	8.23	8.24	6.88	5.75	5.22	5.2	4.41	6.23
Other RD *n* (%)	2314	2408	2494	2718	2431	2556	2272	17,193
H33.5	24.34	25.25	23.62	25.84	24.14	24.32	21.91	24.19
Unspecified *n* (%)	1919	1626	2967	2728	2636	2592	2876	17,344
H33	20.19	17.05	28.1	25.93	26.18	24.66	27.74	24.40
Repair of retinal tear by diathermy, cryotherapy or laser photocoagulation	479	405	369	317	232	304	241	2347
14.3 with extensions *n* (%)	5.04	4.25	3.49	3.01	2.30	2.89	2.32	3.30
Repair of retinal detachment with scleral buckling and implant	1055	980	775	777	607	585	602	5381
14.4 with extensions *n* (%)	11.10	10.27	7.34	7.39	6.03	5.57	5.81	7.57
Repair of retinal detachment with diathermy, cryotherapy or laser photocoagulation	340	332	238	205	234	219	215	1783
14.5 with extensions *n* (%)	3.58	3.48	2.25	1.95	2.32	2.08	2.07	2.51
Repair of retinal detachment with ppv vitrectomy (and injection of vitreous substitute)	4312	4751	4747	5019	5123	5652	5792	35,396
14.74, 14.75 *n* (%)	45.36	49.81	44.95	47.71	50.88	53.78	55.86	49.80
Other repair of retinal detachment	73	37	27	17	11	17	10	192
14.9 *n* (%)	0.77	0.39	0.26	0.16	0.11	0.16	0.10	0.27

**Table 4 jcm-12-01461-t004:** Multiple logistic regression models of the risk factors for different types of RD among Polish adults from 2013 to 2019.

	RD with Retinal Break				Traction RD				Serous RD			
	OR	2.5% OR	97.5% OR	*p*-Value	OR	2.5% OR	97.5% OR	*p*-Value	OR	2.5% OR	97.5% OR	*p*-Value
Age	1.026	1.025	1.027	<0.001	1.013	1.012	1.015	<0.001	1.007	1.005	1.008	<0.001
Male sex	2.32	2.265	2.376	<0.001	2.785	2.638	2.94	<0.001	2.335	2.245	2.43	<0.001
Rural residence	0.958	0.932	0.984	0.002	1.06	0.999	1.124	0.054	0.765	0.73	0.801	<0.001
DM type 1 E10	0.971	0.922	1.023	0.277	0.99	0.961	1.035	0.65	2.933	2.757	3.121	<0.001
DM type 2 E11	1.603	1.545	1.663	<0.001	1.01	0.956	1.065	0.488	0.984	0.932	1.039	0.567
DR H36.0	2.109	1.973	2.254	<0.001	2.493	2.143	2.889	<0.001	7.046	6.602	7.521	<0.001
Myopia H44.2 and H52.1	2.997	2.89	3.107	<0.001	2.255	2.082	2.44	<0.001	1.972	1.85	2.101	<0.001
Glaucoma H40	2.168	2.113	2.224	<0.001	1.904	1.795	2.02	<0.001	3.089	2.965	3.219	<0.001
Uveitis H20	2.561	2.375	2.762	<0.001	4.214	3.725	4.754	<0.001	4.46	4.051	4.907	<0.001

## Data Availability

The National Health Fund Registry data are available at http://www.nfz.gov.pl (accessed 1 October 2022) and the Statistics Poland data are available at http://www.stat.gov.pl (accessed 1 October 2022).

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
