# Peer review of "First Nation-Wide Study of the Incidence and Characteristics of Retinal Detachment in Poland during 2013–2019"

_jcm, 2023, doi:10.3390/jcm12041461_

Round 1

Reviewer 1 Report

The authors reported the incidence and characteristics of retinal detachment in Poland during 2013-2019. However, there are some questions in the manuscript.

1.Abstract, Line 20: "........RD treatment procedures including pars plana vitrectomy (PPV), scleral buckling and implant". Is "implant" the third method to treat RD? Please explain what does "implant" mean? Besides, some retinal specialists can also use gas tamponade (SF6 or C3F8) to treat rhegmatogenous RD. Why is gas tamponade not included in this study?

2. Abstract, Line 22: "The average incidence was 32.6/100,000 person-years........ The overall incidences of rhegmatogenous RD, traction RD and serous RD were 13.72/100,000 , 2.03/100,000 and 1.02/100,000 person years, respectively." It looks weird since "32.6" is much higher than "13.72+2.03+1.02". Please explain the reason.

3. Table 1. Age-standardized incidence of RD among Polish adults from 2013 to 2019 by age group. There is no data of No. age <18 years. Why?

4. Results, Line 161: "Multivariate logistic regression models were constructed to analyze the risk factors for rhegmatogenous RD (with retinal break), traction and serous RD in Poland during 2010-2015." However, this study is during 2013-2019.

5. Table 1: "Overall incidence/1,000,000 person-yrs – (95% CI: ) 326.4 (95%CI: 312.79 – 339.99); Incidence in females/1,000,000 person-yrs – (95% CI: ) 326.14 (95%CI: 311.14 – 341.13); Incidence in males/1,000,000 person-yrs – (95% CI: ) 326.67 (95%CI: 311.57 – 341.23)" The incidence of RD in females and males looks very similar. However, in Line164: ".....age (OR 1.026), male sex (OR 2.320)...." Please explain the reason why the age has so low OR and the male sex has so high OR.

6. Discussion, Line 181: "The other RD (H33.5) and unspecified RD (H33) cases were diagnosed in 24.19% and 24.40% of patients, which may have biased the findings of specific subtypes." Please explain what are the other RD and unspecified RD? Is it possible that these two codes (H33.5 and H33) are wrong coding and they should be excluded to avoid bias?

Author Response

Dear Editors, Dear Reviewers,

 We would like to thank you and other reviewers for your kind, friendly and instructive comments on our paper. Following your suggestions we have made some corrections. Please find my responses and list of the changes along with modified manuscript as an attached file. 

Yours sincerely

                                                                                 Michal S. Nowak MD, PhD – corresponding author  Reviewer 1

The authors reported the incidence and characteristics of retinal detachment in Poland during 2013-2019. However, there are some questions in the manuscript.

  1. Abstract, Line 20: "........RD treatment procedures including pars plana vitrectomy (PPV), scleral buckling and implant". Is "implant" the third method to treat RD? Please explain what does "implant" mean? Besides, some retinal specialists can also use gas tamponade (SF6 or C3F8) to treat rhegmatogenous RD. Why is gas tamponade not included in this study?

Ad 1. Yes, we agree with this comment. This could have made some confusions. However, the definition of ICD-9 codes 14.4 with extensions is “repair of retinal detachment with scleral buckling and implant”. The implant is a seal. This definition is already mentioned in the materials and methods section and we have removed it from abstract. The gas tamponade is enclosed in “other treatment methods” of RD.

  1. Abstract, Line 22: "The average incidence was 32.6/100,000 person-years........ The overall incidences of rhegmatogenous RD, traction RD and serous RD were 13.72/100,000 , 2.03/100,000 and 1.02/100,000 person years, respectively." It looks weird since "32.6" is much higher than "13.72+2.03+1.02". Please explain the reason.

Ad 2. Yes, we agree with this comment. This could have made some confusions. We have added the missing values: The overall incidences of rhegmatogenous RD, traction RD, serous RD, other RD and unspecified RD were 13.72/100,000 , 2.03/100,000 , 1.02/100,000 , 7.90/100,000 and 7.97/100,000 person-years, respectively. Please see, in the modified manuscript.

  1. Table 1. Age-standardized incidence of RD among Polish adults from 2013 to 2019 by age group. There is no data of No. age <18 years. Why?

Ad 3. Yes, we agree with this comment. This could have made some confusions. Our analysis included only adult patients and this was added into the Abstract, Aim, Materials and methods sections. Please see, in the modified manuscript.

  1. Results, Line 161: "Multivariate logistic regression models were constructed to analyze the risk factors for rhegmatogenous RD (with retinal break), traction and serous RD in Poland during 2010-2015." However, this study is during 2013-2019.

Ad 4. Yes, we agree with this comment. We have changed it. Please see, in the modified manuscript.

  1. Table 1: "Overall incidence/1,000,000 person-yrs – (95% CI: ) 326.4 (95%CI: 312.79 – 339.99); Incidence in females/1,000,000 person-yrs – (95% CI: ) 326.14 (95%CI: 311.14 – 341.13); Incidence in males/1,000,000 person-yrs – (95% CI: ) 326.67 (95%CI: 311.57 – 341.23)" The incidence of RD in females and males looks very similar. However, in Line164: ".....age (OR 1.026), male sex (OR 2.320)...." Please explain the reason why the age has so low OR and the male sex has so high OR.

Ad 5. Yes, we agree with this comment. This could have made some confusions. The age is continuous variable (risk increases or decreases by every year of age) and usually the OR is smaller when compared with non-continuous variables. In our study population the sex distribution differed between RD subtypes. Additionally, most studies indicated a preponderance of men regarding the incidence of rhegmatogenous RD; our study demonstrated similar results with the OR of 2.32 (95% CI: 2.265 to 2.376; p<0.001) in male sex. Men are at higher risk for ocular trauma, which could contribute either acutely or remotely rhegmatogenous RD formation. However, male preponderance exists even after exclusion of trauma and pseudophakic RD which is more common in women. This is also presumably associated with the fact that males have a longer axial length, more posteriorly located vitreous base and are more affected by posterior vitreous detachment-associated rhegmatogenous RD. This information was added into the results and discussion sections. Please see, in the modified manuscript.

  1. Discussion, Line 181: "The other RD (H33.5) and unspecified RD (H33) cases were diagnosed in 24.19% and 24.40% of patients, which may have biased the findings of specific subtypes." Please explain what are the other RD and unspecified RD? Is it possible that these two codes (H33.5 and H33) are wrong coding and they should be excluded to avoid bias?

Ad 6. Yes, we partly agree with this comment. This could have made some confusions. However, all included subjects were identified by matching personal identification number (PESEL) with the ICD-10 codes (H33.0, H33.2, H33.4, H33.5 and H33 (unspecified RD)) so there is no bias. But some of specific codes (H33.0, H33.2, H33.4) could have been coded as H33.5 or H33 and we are not able to verify these. This was explained in the discussion section. Please see in the modified manuscript.

We have also made some changes required by other reviewers. Please see, in the modified manuscript.

Reviewer 2 Report

The authors constructed a well-designed nationwide study of the incidence and characteristics of retinal detachment in Poland during 2013-2019. Their results provided novel insights into retinal detachment management. Some revisions are required.

Some sentences in the Abstract could be more concise.

The number format needs to be unified (100 000 or 100,000).

I suggest the authors list the diagnostic coding involved in this study in a table.

In Figure 1, I think the H33 means H33 with subtypes.

The authors should detail the control group selection.

The authors appear to have not fully reported statistical results, as described in the Methods section.

The authors are suggested to further discuss the underlying mechanism of the risk factors. E.g., As the results show, DM type I significantly influenced serous RD, and DM type II affected rhegmatogenous RD.

The Conclusions section needs to be reconstructed and enriched.

Author Response

Dear Editors, Dear Reviewers,

 We would like to thank you and other reviewers for your kind, friendly and instructive comments on our paper. Following your suggestions we have made some corrections. Please find my responses and list of the changes along with modified manuscript as an attached file. 

Yours sincerely

                                                                                 Michal S. Nowak MD, PhD – corresponding author  

Reviewer 2

The authors constructed a well-designed nationwide study of the incidence and characteristics of retinal detachment in Poland during 2013-2019. Their results provided novel insights into retinal detachment management. Some revisions are required.

  1. Some sentences in the Abstract could be more concise.

Ad 1. Yes, we agree with this comment. We have rewritten the abstract. Please see, in the modified manuscript.

  1. The number format needs to be unified (100 000 or 100,000).

Ad 2. Yes, we agree with this comment. We have unified number format into 100,000 person-years in the whole text. Please see, in the modified manuscript.

  1. I suggest the authors list the diagnostic coding involved in this study in a table.

Ad 3. Yes, we partly agree with this comment. We have checked that all diagnostic codes of RDs have been already mentioned in materials and methods section i.e. H33.0, H33.2, H33.4, H33.5 and H33 (unspecified RD).

  1. In Figure 1, I think the H33 means H33 with subtypes.

Ad 4. Yes, we agree with this comment and we have added new Figure 1.

  1. The authors should detail the control group selection.

Ad 5. Yes, we agree with this comment. For regression analysis, the control group of the same size was randomly selected from the NHF database from patients treated in 2013-2019 who had not been diagnosed with RD. It was added into the materials and methods section. Please see, in the modified manuscript.

  1. The authors appear to have not fully reported statistical results, as described in the Methods section.

Ad 6. Thank you for this suggestion. We have checked that all analyses mentioned in materials and methods section have been already presented in the results section.

  1. The authors are suggested to further discuss the underlying mechanism of the risk factors. E.g., As the results show, DM type I significantly influenced serous RD, and DM type II affected rhegmatogenous RD.

Ad 7. Yes, we agree with this comment. The discussion has been significantly expanded and we have also added another 19 references. Please see the discussion section in the modified manuscript.

  1. The Conclusions section needs to be reconstructed and enriched.

Ad 8. Yes, we agree with this comment. The conclusions have been rewritten. Please see the conclusions section in the modified manuscript.

We have also made some changes required by other reviewers. Please see, in the modified manuscript.

Round 2

Reviewer 1 Report

The manuscript looks better after revision. However, there are still some questions:

1.  Introduction, Line 50: "Moreover, nontraumatic phakic detachments had the highest demand incidence of 9.7 per 100,000, while nontraumatic pseudophakic and aphakic RD were significantly less common, at 1.2 and 0.3 per 100,000, respectively." Are phakic eyes have highest incidence of RD? Are pseudophakia and aphakia risk factors for retinal detachment in this study? Besides, "demand" seems unnecessary in the sentence.

2. 2.2. Statistical analyses, Line 99: "The second part of the present study focused of clinical and treatment characteristics of RD and on risk factor analysis." What does "focus of" mean? Please rewrite this sentence.

3. 3. Results: Line 164: "Our analysis showed that rhegmatogenous RD was significantly associated with age (OR 1.026), male sex (OR 2.320) ........." However, in the end of Table 1: "Overall incidence/100,000 person-yrs – (95% CI: ) 32.64 (95%CI: 31.28 – 33.99); Incidence in females/100,000 person-yrs – (95% CI: ) 32.61 (95%CI: 31.11 – 34.11); Incidence in males/100,000 person-yrs – (95% CI: ) 32.67 (95%CI: 31.16 – 34.12). The incidence of RD in both sexes looks very similar. How can male sex have so high OR (2.320)?

4. 4. Discussion, Line 237: "..... in which which we did not find...." There are two "which" in this sentence. Please correct it.

Author Response

Dear Editors, Dear Reviewers,

 We would like to thank you and other reviewers for your kind, friendly and instructive comments on our paper. Following your suggestions we have made some corrections. Please find my responses and list of the changes along with modified manuscript as an attached file. 

Yours sincerely

                                                                                 Michal S. Nowak MD, PhD – corresponding author  Reviewer 1

The manuscript looks better after revision. However, there are still some questions:

  1. Introduction, Line 50: "Moreover, nontraumatic phakic detachments had the highest demand incidence of 9.7 per 100,000, while nontraumatic pseudophakic and aphakic RD were significantly less common, at 1.2 and 0.3 per 100,000, respectively." Are phakic eyes have highest incidence of RD? Are pseudophakia and aphakia risk factors for retinal detachment in this study? Besides, "demand" seems unnecessary in the sentence.

Ad 1. Yes, we agree with this comment. This could have made some confusions. This information in the introduction (line 50) refers to the incidence of retinal detachments in two districts in the West Midlands (UK). We have added the citation to this sentence. We have also removed the word “demand”. Please see, in the modified manuscript. The pseudophakia and aphakia were not included into the regression analysis in the present study.

  1. Statistical analyses, Line 99: "The second part of the present study focused of clinical and treatment characteristics of RD and on risk factor analysis." What does "focus of" mean? Please rewrite this sentence.

Ad 2. Yes, we agree with this comment. This could have made some confusions. We have rewritten this sentence: “The second part of the present study included clinical and treatment characteristics of RD and risk factor analysis”. Please see, in the modified manuscript.

  1. Results: Line 164: "Our analysis showed that rhegmatogenous RD was significantly associated with age (OR 1.026), male sex (OR 2.320) ........." However, in the end of Table 1: "Overall incidence/100,000 person-yrs – (95% CI: ) 32.64 (95%CI: 31.28 – 33.99); Incidence in females/100,000 person-yrs – (95% CI: ) 32.61 (95%CI: 31.11 – 34.11); Incidence in males/100,000 person-yrs – (95% CI: ) 32.67 (95%CI: 31.16 – 34.12). The incidence of RD in both sexes looks very similar. How can male sex have so high OR (2.320)?

Ad 3. Yes, we agree with this comment. This could have made some confusions. In our study population the sex distribution differed between RD subtypes. In subjects with rhegmatogenous RD there was 51.47% of men and 48.53% of women. This information is already included in the results section, lines 116-188: “The incidence was slightly higher in men when compared to women and the sex distribution differed between RD subtypes.” Additionally, most studies indicated a preponderance of men regarding the incidence of rhegmatogenous RD; our study demonstrated similar results with the OR of 2.32 (95% CI: 2.265 to 2.376; p<0.001) in male sex. This information is already included in the discussion section, lines 242-245: “Most studies indicated a preponderance of men regarding the incidence of rhegmatogenous RD; our study demonstrated similar results with the OR of 2.32 (95% CI: 2.265 to 2.376; p<0.001) in male sex.”

  1. Discussion, Line 237: "..... in which which we did not find...." There are two "which" in this sentence. Please correct it.

Ad 4. Yes, we agree with this comment. We have corrected it. Please see, in the modified manuscript.

Reviewer 2 Report

The authors revised the manuscript carefully and adequately. The sentence (Line 237) needs to be revised. 

Author Response

Reviewer 2

  1. The authors revised the manuscript carefully and adequately. The sentence (Line 237) needs to be revised. 

Ad 1. Thank you very much for this comment. We have corrected line 237. Please see, in the modified manuscript.

We have also made some changes required by other reviewers. Please see, in the modified manuscript.
